# Fruit Morphology and Ripening-Related QTLs in a Newly Developed Introgression Line Collection of the Elite Varieties ‘Védrantais’ and ‘Piel de Sapo’

**DOI:** 10.3390/plants11223120

**Published:** 2022-11-15

**Authors:** Miguel Santo Domingo, Carlos Mayobre, Lara Pereira, Jason Argyris, Laura Valverde, Ana Montserrat Martín-Hernández, Jordi Garcia-Mas, Marta Pujol

**Affiliations:** 1Centre for Research in Agricultural Genomics (CRAG), CSIC-IRTA-UAB-UB, Edifici CRAG, Campus UAB, 08193 Barcelona, Spain; 2Institut de Recerca i Tecnologia Agroalimentàries (IRTA), 08193 Barcelona, Spain

**Keywords:** introgression lines, *Cucumis melo*, fruit quality, climacteric ripening

## Abstract

Melon is an economically important crop with widely diverse fruit morphology and ripening characteristics. Its diploid sequenced genome and multiple genomic tools make this species suitable to study the genetic architecture of fruit traits. With the development of this introgression line population of the elite varieties ‘Piel de Sapo’ and ‘Védrantais’, we present a powerful tool to study fruit morphology and ripening traits that can also facilitate characterization or pyramidation of QTLs in *inodorous* melon types. The population consists of 36 lines covering almost 98% of the melon genome, with an average of three introgressions per chromosome and segregating for multiple fruit traits: morphology, ripening and quality. High variability in fruit morphology was found within the population, with 24 QTLs affecting six different traits, confirming previously reported QTLs and two newly detected QTLs, *FLQW5.1* and *FWQW7.1*. We detected 20 QTLs affecting fruit ripening traits, six of them reported for the first time, two affecting the timing of yellowing of the rind (*EYELLQW1.1* and *EYELLQW8.1*) and four at the end of chromosome 8 affecting aroma, abscission and harvest date (*EAROQW8.3*, *EALFQW8.3*, *ABSQW8.3* and *HARQW8.3*). We also confirmed the location of several QTLs, such as fruit-quality-related QTLs affecting rind and flesh appearance and flesh firmness.

## 1. Introduction

Cultivated fruits and vegetables are an essential part of the human and animal diet, so it is crucial to improve the yield as well as the quality of their edible parts. To help breeders, molecular tools have been developed in many crop species to map and associate a trait with a genomic region. With these tools, molecular plant breeding has emerged as a relevant strategy to accelerate plant breeding, and the use of molecular markers has led to the discovery of molecular mechanisms of diverse traits [1,2,3].

Several mapping populations have been used in modern plant breeding, such as recombinant inbred lines (RILs) [4,5,6] and F2 populations [7,8]. RILs have been widely used for mapping QTLs associated with traits such as yield in wheat [9] and leaf type in soybean [10], while F2 populations have also been used in the study of yield in rice [11] and fruit quality traits in pomegranate [12]. In RILs and F2 populations, the genome of the two founding parents is equally represented, but, in other mapping populations, the genomes of the parents are not equally represented, such as introgression lines (ILs) [13,14] and chromosome segment substitution lines (CSSLs) [15,16]. 

IL populations make it possible to mendelize and study different traits without the effect of other interacting loci. This has been used widely for different traits and crops, such as fruit weight and soluble solid content in tomato [13], flowering time in maize [17], grain weight in rice [18] or CMV resistance in melon [19,20]. In addition, it allows pyramiding of different QTLs and studying their interactions, as has been conducted for studying resistance to bacterial blight in rice [21] or resistance to *Aphanomyces euteiches* in pea [22]. ILs are developed by multiple backcrosses with the same parental line (the recurrent parent) and a last step of self-pollination to fix the heterozygous regions, providing a set of perpetual lines sharing most of the genome of the recurrent parent but having an introgression in a given chromosome from the other parental line (donor parent). Ideally, with a complete set of ILs, the donor genome should be fully covered. 

In melon, many mapping populations have been developed to study fruit quality [23,24,25,26,27] and fruit morphology [27,28,29]. For melon breeding, fruit ripening represents a key complex trait, and several studies have been conducted to identify the genes underlying this process [27,30,31]. Melon presents intra-specific segregation of climacteric ripening, allowing the development of populations segregating for ripening behavior [27,30]. Recently, three genes involved in melon ripening (*CmCTR1, CmROS1* and *CmNAC-NOR*) have been validated through CRISPR/Cas9 [32,33], and a collection of lines pyramided with three ripening-related QTLs introgressed in a non-climacteric background has been developed to understand the interaction of these QTLs [34]. 

Here, we developed a novel IL population with ‘Piel de Sapo’ (PS) as the recurrent parent and ‘Védrantais’ (Ved) as the donor parent. These two elite cultivars were chosen due to their differences in fruit morphology and ripening behavior. This is a new high-value genomic resource for breeding *inodorous* varieties and offers some insights into the genomic architecture of fruit quality, fruit morphology and fruit ripening behavior in melon. 

## 2. Results and Discussion

### 2.1. Phenotyping of the Parental Lines

PS and Ved were used as parental lines for the population, with PS the recurrent and Ved the donor parent. Both varieties are considered elite cultivars as they are widely consumed for their taste and aroma but with distinctive characteristics.

Over the two-year period of analysis, 2020 and 2021, PS produced consistently larger fruits than Ved, even though the PS fruits were smaller in the 2021 season (FW; Figure 1A; Appendix A). The soluble solid content was highly dependent on the environmental conditions. In 2020, there were no significant differences between the cultivars, while PS fruits accumulated more soluble solids than Ved in 2021 (SSC; Figure 1B; Appendix A). This SSC variability has been observed in previous studies: the mean value in PS is generally around 10–11° Brix, but, in some environments, it can reach values lower than 9 or higher than 13 [26,27,35,36]. In Ved, the variation has also been found to be highly dependent on the location and the season, from around 7 to 13° Brix [27,31,34]. This high dependence on the environment can make it difficult to map QTLs related with this trait. To overcome this problem, the development of ILs that facilitate the analysis of many replicates in different years and locations and without the interaction among loci observed in F2 and RIL populations is essential. In this way, QTLs consistently observed under different environmental conditions will be detected and later used for specific breeding programs.

No significant differences were found in firmness of the flesh between cultivars or between years (FIR; Figure 1C; Appendix A). This trait is also highly dependent on the environment, especially in PS, where it has been found to vary significantly between years, while Ved tends to be more stable [30,34].

Concerning fruit shape, PS produced more oval melons than Ved in both years, with longer and wider fruits in PS, while Ved produced rounded (FS close to 1) and smaller fruits (FL, FWI, FS; Figure 1D–F; Appendix A).

Significant differences between the parental lines were observed in traits related to ripening behavior. In climacteric melons, production of ethylene is coupled to certain ripening-related traits, such as production of aroma (EARO) and abscission layer formation (EALF), also affecting harvest date (HAR) [27]. While Ved ripened at around 35 days after pollination in both years, PS ripened around 20 days later, generally without producing aroma or abscission layer (Figure 1G–I) (Appendix A). The ripening behavior observed for both parental lines was in agreement with previous studies [27,30,33]. 

### 2.2. Development of the IL Collection

For the development of the IL population, we used the same sets of SNPs as in Pereira et al. 2021 [27]. Starting the Ved × PS cross in 2014, six or seven generations were needed to complete the population, depending on the IL (Figure 2). 

BC1 generation carried a mean of 50.2% of the PS genome, ranging from 10.6% to 76.6%, and an average of 11.8 introgressions, varying between 5 and 17 introgressions per plant. Twenty BC1 plants with a mean of 8.3 introgressions and 65.6% of the recurrent parent were selected for the next round of backcrossing. In BC2 generation, 46 seedlings carrying a mean of 5.3 introgressions and 77.3% of the PS genome were selected for the next round of backcrossing. In BC3 generation, 55 plants carrying an average of 3.4 introgressions and 82.4% of the PS genome were selected. Some BC3 plants were backcrossed, while others were self-pollinated. During the spring season of 2017, both BC3S1 and BC4 generations were genotyped, and 92 plants with an average of 2.2 introgressions and 89.7% of the PS genome were selected. Seven BC3 families were also recovered in order to cover gaps in the genome. At this point, five lines carried the desired single introgression. To complete the population, in summer 2017, several interesting progenies were genotyped using the complete set of SNPs or a customized number of SNPs, depending on the line. With this new screening, 18 new ILs were obtained. In 2018, 24 additional families were genotyped and 29 ILs were finalized (considering the ILs obtained in 2017). Moreover, in 2018, three more families were genotyped with the complete set of SNPs to cover gaps in the genome. Finally, five ILs were added to the population in 2019, and two more in 2020, which were derived from the lines selected for covering gaps or unfinished lines. 

In the BC1 generation, the mean heterozygosity was H0 = 0.48, and segregation distortion was observed for 10 SNPs. The highest distortion, χ^2^ = 18, was in the SNP chr7_16723157. The H0 of this SNP, located at the position 18,255,891 bp on chromosome 7, was 0.63, with a predominance of the Ved allele. Segregation distortion has previously been observed in melon in different chromosomes, including chromosome 7 [37]. For the other SNPs that had segregation distortion, the H0 ranged from 0.41 to 0.44, below the mean. 

Finally, a set of 36 ILs was developed, covering 97.89% of the genome, with an average of three unique introgressions per chromosome (Appendix A). The information and graphical representation of the population are provided in Table 1 and Figure 3. The developed IL population had a mean of three introgressions per chromosome, with a range between two and four, and an average size of 14.6 Mb per chromosome. Both number of introgressions and average size are similar to previous IL populations developed in melon [14,27,28].

### 2.3. Phenotyping of the IL Population and QTL Mapping

In 2019, a first phenotypic analysis of a subgroup of 17 ILs was performed (Appendix A). In summer 2020, the available collection of 35 ILs was properly characterized and phenotyped (Appendix A) and the population was genotyped using the complete set of SNPs described by Pereira et al. 2021 [27] (Appendix A). Finally, in 2021, six unfinished lines in 2020 were phenotyped again, together with a newly developed line covering a gap on chromosome 5 (Appendix A). 

#### 2.3.1. Fruit Quality Traits

During the phenotyping, the following fruit quality traits were analyzed: mottled rind (MOT), immature rind color (ECOL), yellowing of the rind (YELL), presence of sutures (SUT), flesh color (FC), soluble solid content (SSC) and flesh firmness (FIR). Those related with the external or internal appearance of the fruit were treated as qualitative (Figure 4A–E). The SSC and FIR, as well as morphological and ripening-associated traits, were treated as quantitative. The QTLs for quantitative traits are shown in Table 2.

Mottled rind is a characteristic of the parental line PS. Within the IL collection, only VED2.3 did not have mottled rind (Figure 4A). This QTL was mapped at the end of chromosome 2, at interval 20.37–24.77 Mb. A major gene, *Mt-2*, has been previously described close to this region [23,27,38], and a candidate gene, *MELO3C026282.2*, encoding a protein essential for chloroplast development, has recently been reported as the causal gene of this QTL [39].

Both VED7.1 and VED7.2 had white immature rind compared to the green rind of PS and the rest of the ILs. This QTL was mapped to interval 2.62–20.73 Mb on chromosome 7 (Figure 4B). This region includes the major gene *Wi*, previously described as controlling this trait [40]. This interval colocalized with the region identified in two other populations of the same parental lines [23,27]. 

Related to the external color of the fruit as well, the entire population turned yellow during ripening (from around 25 DAP), except for VED10.1 and VED10.2, which remained dark green until harvest. This behavior has been related to flavonoid biosynthesis [41]. This trait was mapped on chromosome 10, in interval 1.76–5.11 Mb (Figure 4C), colocalizing with the previously described causal gene *CmKBF* [41].

Regarding the external aspect of the rind, VED11.1 and VED11.2 had sutures, while the rest of the population had smooth rind. This trait was mapped to a region on chromosome 11, in interval 14.14–29.79 Mb (Figure 4D). This region colocalized with QTLs reported in previous studies [23,27] and contains *MELO3C019694.2*, a gene that has been proposed as a candidate gene for this trait [42].

As for the internal appearance of the melon, we detected three different flesh colors (Figure 3B). While PS and most of the population had white flesh, other ILs had orange or green flesh. The flesh of VED8.3 was green when mature, mapping this QTL to chromosome 8 at interval 25.00–32.87. This region colocalizes with previously reported QTLs controlling this trait [43,44] and contains two genes that have been proposed as candidate genes for *Wf*, *MELO3C003069.2* [45] and *MELO3C003097.2* [42]. The orange flesh of VED9.1 and VED9.2 was similar to the Ved parental line, and this QTL was mapped in interval 1.62–22.56 Mb on chromosome 9 (Figure 4E). Gene *CmOr* (*Gf*), located in our mapping interval on chromosome 9, has been reported as controlling accumulation of carotenoids in melon flesh, provoking the orange coloration [46]. The color of the flesh in melon is controlled by two genes under dominant epistasis, being *Gf* dominant over *Wf.* When *Gf* is not present (*gf/gf*), then white is dominant over green [43,44,45]. In our population, PS has the alleles *gf/gf Wf/Wf* and Ved is *Gf/Gf wf/wf*, so VED8.3 had the alleles *gf/gf wf/wf,* resulting in green flesh, and VED9.1 and VED9.2 had *Gf/Gf Wf/Wf*, being orange-fleshed.

Focusing on the other quality traits, we did not detect any significant difference regarding SSC in the population, even though a certain level of variability was observed (Figure 4F) (Appendix A). We did, however, map several QTLs related with fruit firmness (Table 2). The flesh in three of the ILs on chromosome 8 (VED8.1, VED8.2 and VED8.3) was less firm compared to PS (*FIRQW8.1*, *FIRQW8.2* and *FIRQW8.3*), probably caused by their climacteric behavior (Appendix A). The effect of *FIRQV8.1* and *FIRQV8.3* was observed in our first phenotyping in 2019 (Appendix A). Three other ILs (VED2.2, VED2.3 and VED11.2) had an increase in firmness (*FIRQW2.1* and *FIRQW11.1*). In VED2.2 and VED2.3, this was significantly higher than in PS, as was the case for VED2.3 firmness in 2021, although this was not significant. *FIRQW2.1* was mapped to interval 3.16–24.77 on chromosome 2. This QTL was also detected in the reciprocal IL population containing introgressions of PS in the background of Ved [27], with the overlapping region of both populations being 3.16–16.42 Mb. The VED11.2 fruit flesh was significantly firmer than PS, with a QTL (*FIRQW1.1*) on chromosome 11 at interval 29.79–34.46 (Figure 4G, Appendix A), which colocalized with a previously reported flesh firmness QTL [37]. 

#### 2.3.2. Fruit Morphology Traits

Fruit morphology is an important characteristic for fruit consumption, varying from small, round pocket melons to elongated large-sized varieties. Our parental lines had significantly different morphologic phenotypes in both evaluated years (Figure 1D–F), and the IL population segregated for these traits.

On chromosome 7, there was a significant decrease of 50% of fruit weight in VED7.1 and VED7.2 between 2.62 and 20.73 Mb, containing QTL *FWQW7.1* (Appendix A, Figure 5A). We also detected the effect of this QTL in FP, FA, FL and FWI, with a decrease in all these traits (Figure 5, Table 2). In 2019, VED7.1 also produced smaller melons than PS (Appendix A). A consensus QTL in this region has been previously reported, affecting fruit width in a melon RIL population between PI 414723 (subspecies *agrestis*) and ‘Dulce’ (subspecies *melo*), named *fwi7.1* [47]. This QTL was located at 43 cM in chromosome 7; therefore, it should colocalize with both introgressions in VED7.1 and VED7.2, explaining the similar behavior observed in these two lines. However, our QTL was found to affect other traits related with fruit size. Since fruit morphology traits are shown to be correlated in melon [23,48], *fwi7.1* and *FWQW7.1* might be different QTLs, *fwi7.1* being specific to fruit width, while *FWQW7.1* is more general, affecting fruit size. *FWQW7.1* can be useful in melon breeding programs for modifying fruit size without affecting fruit shape. 

A QTL cluster for fruit morphology was detected on chromosome 4, decreasing FP, FA and FWI but not FL, making the fruits thinner (Appendix A). It was mapped to interval 22.48–27.25 Mb (Figure 5, Table 2). QTLs affecting fruit morphology have been previously reported in this chromosome [48,49]. This QTL was not detected in 2021 in VED4.2 (Figure 5). This line contained three SNPs in heterozygosis (SNP 30, 31 and 82 (Appendix A)) in 2020 that were fixed in 2021. It may be possible that this QTL is unstable, or that, in the fixation process, the allele causing the effect in fruit morphology was lost. During the 2021 season, we detected another QTL for FP in the same chromosome but in a different interval (0–14.92 Mb), *FPQW4.2*, and with the opposite effect, producing larger melons. *FPQW4.2* colocalizes with a QTL detected in a reciprocal IL population with the same parental lines, *FPQP4.1* [27]. Both QTLs on chromosome 4, *FPQW4.2* and *FPQP4.1,* consistently affect fruit size, producing larger melons when the Ved allele is present. 

On chromosome 8, we identified two different QTLs for fruit morphology, one in VED8.1 and the other in VED8.3. The QTL in VED8.1 produced flatter melons (Appendix A, Figure 5) and significant changes in FP, FA and FL; it was mapped in interval 0–6.89 Mb at the beginning of the chromosome. This QTL collocated with a previously known QTL for fruit shape in an F2 population with PS as one of the parents [50]. Although not affecting the shape of the fruit, a QTL for fruit weight has also been mapped in the same region, producing smaller melons [23]. The second QTL, affecting VED8.3, was located at 31.12–34.62 Mb (Appendix A, Figure 5) and affects fruit size (FA, FP, FL and FWI), producing smaller melons. This distal part of chromosome 8 has already been reported as one of the main controllers of melon fruit shape [48]. Close to this region, a gene controlling fruit shape has been characterized, *CmOFP13* [29]. Our QTL does not colocalize with this gene, so the causal gene for the QTL covered by VED8.3 must be a different one. Both QTLs on chromosome 8 affecting fruit shape were also detected in 2019 (Appendix A).

A QTL mapped on chromosome 5 in interval 27.63–29.32 Mb (*FLQW5.1*) decreased the fruit length and, consequently, the fruit area (Appendix A, Figure 5). To our knowledge, this is the first time that a QTL controlling fruit morphology has been mapped to the distal part of chromosome 5 in melon (Table 2).

We detected two QTLs affecting fruit shape on chromosome 6 in interval 7.47–35.32 Mb, *FSQV6.1* and *FLQV6.1*. VED6.3 produced more rounded melons than PS, significantly decreasing FL and FS (Appendix A, Figure 5). An association between this region of chromosome 6 and fruit shape has been reported previously in two different RIL populations [23,24] and an F2 population [50]. *FSQV6.1* was also detected in 2019 (Appendix A).

The last fruit-morphology-related QTL (*FSQV11.1*) was detected on chromosome 11 (29.79–34.46 Mb). VED11.2 produced rounder and also smaller melons, affecting FS, FL, FA and FP. This QTL has been previously reported in an RIL population funded by the same parents [23]. 

The presence of QTLs affecting all traits related to fruit morphology, together with other QTLs affecting only some dimensions, brings out the complex genetic architecture of fruit shape and morphology in melon. Some unstable QTLs were detected, such as the QTL affecting size in VED4.2 (*FPQW4.1 and FAQW4.1*). This dependance on the environment in QTL mapping for morphological traits has been reported before in melon when using different populations, such as RILs [23] and ILs [14], and also in tomato [51]. Moreover, in tomato, genetic interactions among OFP, SUN and TRM5 genes have been described, affecting cellular elongation, division and, thus, fruit shape [52]. This variable behavior, together with genetic interactions between QTLs and the large number of genes found within these families in melon, explains the difficulty in controlling those traits in the field. Although many QTLs affecting fruit morphology have already been mapped, the discovery of new QTLs (*FLQW5.1* and *FWQW7.1*) proved the importance of continuing generating genetically diverse material. Unravelling the complete and complex genetic architecture of quantitative traits such as fruit morphology will allow us to better predict the phenotypic performance of melon fruits.

#### 2.3.3. Ripening Traits

The analysis of ripening was centered around four related phenotypes: earliness of aroma production (EARO), earliness of abscission layer formation (EALF), harvest date (HAR) and the level of abscission (ABS). As PS is a non-climacteric melon, most of the fruits did not produce aroma or an abscission layer and were harvested at 55 DAP, around 20 days later than Ved, when they were fully ripe (Figure 1G–I). There were high levels of abscission only in climacteric line Ved, and we detected segregation for these four traits in the IL population, obtaining climacteric ILs.

A QTL on chromosome 6 in the region 7.47–35.32 Mb affected all ripening-related traits. VED6.3 produced climacteric melons, with aroma and an abscission layer around 50 DAP (Appendix A, Figure 6). A QTL related to climacteric ripening has been previously reported in this region in two IL populations. One was with the same parental lines, where the allele of PS delayed ripening in a Ved background [27], and another in an IL population with the same recurrent parent PS, where the introgression of the alleles of PI 161375 induced a climacteric response [53,54]. The causal gene for this QTL has been characterized and identified as transcription factor *CmNAC-NOR* [33,55].

The other chromosome governing climacteric ripening in this IL population was chromosome 8. VED8.1, VED8.2 and VED8.3 had climacteric behavior, similar in VED8.1 and VED8.2, while VED8.3 differed. The climacteric behavior of VED8.1 and VED8.2 was typical, with a sweet aroma and a gradual abscission layer formed between 40 and 45 days (Table 2 and Appendix A; Figure 6). We considered that there were two different QTLs, one in interval 2.63–6.89 Mb and another in interval 6.89–14.98 Mb, due to the existence of a validated QTL in the second region that is not covered by VED8.1, *ETHQV8.1* [30,34]. These two QTLs were also identified in the 2019 season, affecting VED8.1 and VED8.2 (Appendix A). The first QTL, affecting VED8.1, has been previously reported in an RIL population [56]. Affecting VED8.2, the second one was identified in an RIL population with the same parental lines as this study. Widely studied, it has been validated in several seasons both in climacteric and non-climacteric backgrounds [30]. There are three candidates that have been proposed as the causal gene: *MELO3C024516.2*, *MELO3C024518.2* and *MELO3C024520.2*, encoding demethylase, a negative regulator of ethylene signaling, and an ethylene responsive transcription factor, respectively [30]. Two of these candidate genes, *MELO3C024516.2* and *MELO3C024518.2*, have been studied through CRISPR/Cas9 knock-out lines, suggesting involvement of both genes in fruit ripening [32]. A third QTL was mapped on chromosome 8 in interval 31.12–34.62 Mb, affecting VED8.3, VED7.1 and VED7.2. Both VED7.1 and VED7.2 have undesired introgressions on chromosome 8 in the same region as VED8.3 (Figure 3A). These three lines behaved similarly, with abrupt climacteric ripening, extreme flesh softening and very quick abscission: one day after the aroma was detected, the fruits abscised from the plant. This indicates a unique QTL in the three lines, located in the shared region. To our knowledge, this QTL is newly reported as this region was previously reported being involved in flesh firmness, but no effect on climacteric ripening had been described [30,37,53]. This newly described QTL at the end of chromosome 8 (*EAROQW8.3*, *EALFQW8.3*, *ABSQW8.3* and *HARQW8.3*) can be of great importance for breeding since its presence accelerates ripening abruptly. The identification of the genes underlying this QTL and the availability of molecular makers will help breeders to control its presence/absence or to modulate its effect.

The last phenotype related to fruit ripening is the earliness of yellowing of the rind (EYELL). This phenotype is caused by biosynthesis of flavonoids [41]. As shown in Figure 4, there is a major gene controlling this trait on chromosome 10, which we detected in 2020 and 2021 in VED10.1 and VED10.2 (Figure 6B). In the case of VED10.1 in 2020, the region where the gene is located was still segregating, explaining the high variance in this line (Figure 6B). Apart from these two ILs, we detected two more QTLs affecting the timing of the yellowing of the rind (Table 2, Figure 6B). One QTL on chromosome 1, *EYELLQW1.1*, delays this change, while the other, *EYELLQW8.1* on chromosome 8, has a greater effect *(*Table 2). To our knowledge, these are two newly reported QTLs.

## 3. Materials and Methods

### 3.1. Plant Material and Breeding Scheme

Seedlings were planted and maintained for two weeks in biodegradable pots under controlled conditions at CRAG (Barcelona). After genotyping, selected plants were transported to a greenhouse in Caldes de Montbui (Barcelona) and grown in coconut fiber sacks in the spring and summer seasons. These plants were pruned weekly and manually pollinated. Only one fruit per plant was allowed to develop to optimize fruit growth and seed production.

The IL population was created from a cross between two elite commercial cultivars, “Védrantais” (Ved), a French variety belonging to the *cantalupensis* group, and “Piel de Sapo” T111 (PS), a Spanish cultivar in the *inodorous* group. PS is a large, oval, white-fleshed melon with non-climacteric ripening behavior, while Ved is a small, round, aromatic, orange-fleshed melon with typical climacteric behavior. Pollen from male flowers of F1 plants was used to pollinate female flowers from the recurrent parent PS, obtaining BC1 seed, from which the pre-IL female flowers were generally pollinated with pollen from PS flowers. Seedlings of the BC1 progenies were screened and plants were selected following these criteria: (1) those having the complete genome of the donor parent at least twice; (2) the lines carrying the lowest possible number of introgressions in their genome; (3) the lines with the highest possible percentage of the recurrent parent genome. Chosen individuals were backcrossed again with PS to obtain BC2 progeny. Several cycles of genotyping and selection were subsequently carried out for the progenies, as shown in Figure 2**.** For lines with less than three introgressions, self-pollination was used to identify plants carrying a single introgression in homozygosity in their progeny. Depending on the line, three or four backcrosses followed by one or two self-pollinations were carried out in order to obtain the final IL. Lines with undesired introgressions were backcrossed and self-pollinated again.

### 3.2. In Vitro Plant Culture

Selected plantlets were maintained in vitro for several months. Plants were introduced in sterile tubes with modified MS medium (Appendix A). Rooted plantlets were cut and transferred to fresh media every three weeks. Before the acclimatation, plants were multiplied to have enough replicates for the experiment. For the acclimatation, they were transferred to soil in a closed portable greenhouse for two days. Over the following five days, the greenhouse was opened slightly every day until they were totally acclimated.

### 3.3. DNA Extraction and Genotyping

Depending on the objective, we used two different DNA extraction procedures: CTAB protocol for high-throughput genotyping and long-term storage [57] with some modifications [23], and an alkaline-lysis protocol for single-SNP genotyping and short-term storage [58].

Progenies of BC1, BC1 and BC3 were genotyped using a set of 48 SNPs, called set 1 [27]. The selected plants from the BC2 and BC3 generations were then genotyped with an additional set of 48 SNPs (set 2) [27]. SNPs were designed from resequencing data of both parents [59], and their positions relate to the melon reference genome v3.6.1. The progenies of BC3S1, BC4, BC3S2 and BC4S1, screened in 2017, were genotyped with the complete set of 96 SNPs [27]. Seedlings were genotyped with customized sets of SNPs using two similar systems: (i) KASPar SNP Genotyping System (LGC, Teddington, UK), and (ii) PACE2.0 SNP Genotyping System (3CR Bioscience, Harlow, UK). Primers were designed for both systems following the LGC Genomics protocol. High-throughput genotyping used the Biomark HD genotyping system, based on Fluidigm technology. In the last phases of genotyping, only SNPs within the known introgressions were genotyped. The same primers were used for qPCR in a LightCycler 480 Real-time PCR System, according to the technical instructions provided by the supplier (Roche Diagnostics, Barcelona, Spain).

The size of the introgressions was calculated following two assumptions: (i) the haplotypes of the non-genotyped extremes of the chromosome are the same as the first or last genotyped SNP, and (ii) the recombination point is in the intermediate position between two genotyped SNPs. The approximate genetic size of the introgressions was calculated using as a reference the Ved × PS RIL population genetic map [23].

### 3.4. Experimental Design and Phenotyping

The major part of the IL population was phenotyped in the summer of 2020 in Caldes de Montbui (Barcelona), with additional phenotyping of some ILs in the summer of 2019 and 2021. During the three seasons, plants were randomly planted, with at least eight replicates per IL, and 40 PS plants. More than five fruits were characterized per line.

The phenotypes were organized in three categories of trait: fruit quality, fruit morphology and fruit ripening (Table 3)

Fruit quality traits were separated between qualitative and quantitative. Qualitative traits were evaluated visually: mottled rind (MOT), immature rind color (ECOL), yellowing of the rind (YELL), presence of sutures (SUT) and flesh color (FC) (Table 3). Two additional phenotypes related with quality were treated as quantitative: soluble solid content (SSC) and flesh firmness (FIR) (Table 3). SSC was measured at harvest with an optical refractometer from manually extracted juice from at least three 1 cm cylinders of fruit flesh. FIR was also measured at harvest using a penetrometer. Four measurements were recorded per sample, and the average value was used for the analysis.

For fruit morphology traits, scanned images of the fruits were analyzed with the Tomato analyzer software [60], and five phenotypes were annotated: fruit area (FA), fruit perimeter (FP), fruit length (FL), fruit width (FW) and fruit shape (FS), estimated as the ratio of fruit length to fruit width. Fruit weight (FW) was also measured at harvest.

For fruit ripening traits, fruits were examined daily from 20 days after pollination (DAP) until harvest date. Presence of aroma (EARO) was recorded as the first day aroma could be detected by smelling the fruit. Abscission layer formation (EALF) was recorded as the first day the abscission layer was detected by visual inspection. The earliness of the yellowing of the rind (EYELL) was recorded as the first day the bottom of the fruit started to turn yellow. Abscission level (ABS) was treated as semi-quantitative using a scale from 0 to 3, 0 being the total absence of the layer and 3 complete abscission from the plant. Harvest date (HAR) was fixed using the following criteria: (i) for fruits with no abscission layer formation, HAR was fixed at 56 DAP, (ii) for fruits that completely abscised from the plant, HAR was the abscission day and (iii) for fruits with partial abscission layer formation, HAR was fixed at five days after the abscission layer was detected visually. EARO and EALF, if not present, were considered as the same day as HAR.

### 3.5. QTL Mapping and Data Analysis

QTL mapping was performed with a non-parametrical test due to the non-normal distribution of data. Wilcoxon signed-rank test was used to compare each IL with PS, and then a Holm method was used to correct *p*-values for multiple comparisons. Significant QTLs were fixed at *p*-value < 0.05.

All the statistical analyses and graphical representations were obtained using the software R v3.5.3 with the RStudio v1.0.143 interface [61]. 

## 4. Conclusions

The existence of different types of mapping populations allows scientists and breeders to dissect the genetic architecture of complex traits, such as fruit morphology and fruit ripening. With the development of this IL population, we have validated known QTLs and discovered new ones implicated in a diverse set of qualitative and quantitative traits that can be useful in breeding programs for fruit quality, morphology and ripening. IL populations are also a powerful tool to fine-map QTLs and identify candidate genes as a previous step to develop new varieties using marker-assisted selection to introgress natural alleles in elite cultivars or gene-editing to generate new alleles. This IL population and the reciprocal IL population previously obtained [27] represent a good source of allelic variability that can be further used in breeding programs of both *inodorus* and *cantalupensis* melon types. For quality traits, PS cultivars without the characteristic mottled skin, with orange/green flesh or presenting sutures in the rind were obtained. As for fruit size and shape, several QTLs have been detected that may be used to modify fruit size and shape. Regarding fruit ripening, this IL population confirms the key role of *CmNAC-NOR* and *ETHQV8.1* as regulators of this important trait, with the possibility of converting a non-climacteric melon into a climacteric one. The QTLs identified in this work can be further explored in the melon germplasm in search of new alleles that will enrich the breeders’ toolkit.

## Figures and Tables

**Figure 1 plants-11-03120-f001:**
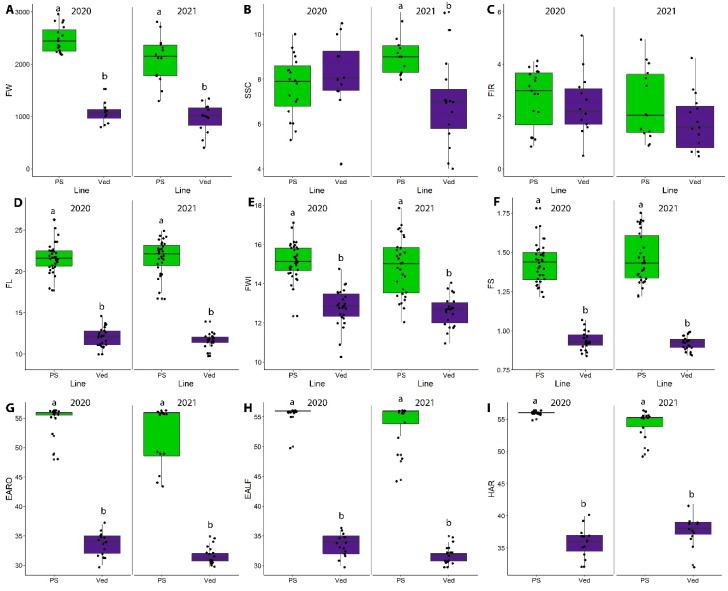
Fruit-quality-, fruit-morphology- and fruit-ripening-related phenotypes of the parental lines PS and Ved in 2020 and 2021. (**A**) Fruit weight (FW) (g). (**B**) Soluble solid content (°Brix) (SSC). (**C**) Flesh firmness (kg cm^−2^) (FIR). (**D**) Fruit length (cm) (FL). (**E**) Fruit width (cm) (FWI). (**F**) Fruit shape (FS). (**G**) Presence of aroma (days after pollination) (EARO). (**H**) Abscission layer formation (days after pollination) (EALF). (**I**) Harvest date (days after pollination) (HAR). Replicates are represented with dark circles. Different letters (a, b) represent significantly different groups (*p*-value < 0.05).

**Figure 2 plants-11-03120-f002:**
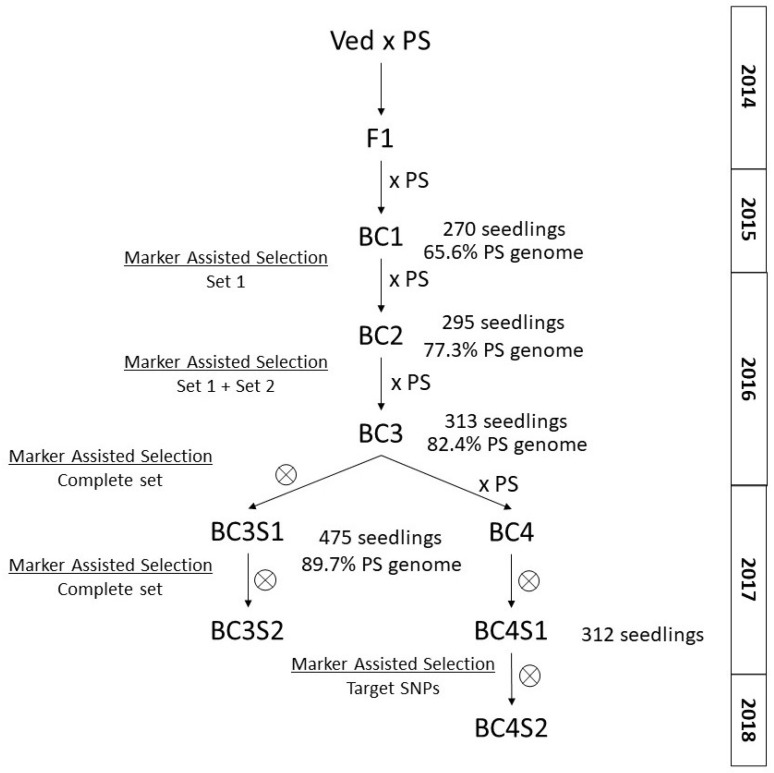
Simplified breeding scheme for the development of the IL population, showing the number of seedlings analyzed and the average PS genome at each generation.

**Figure 3 plants-11-03120-f003:**
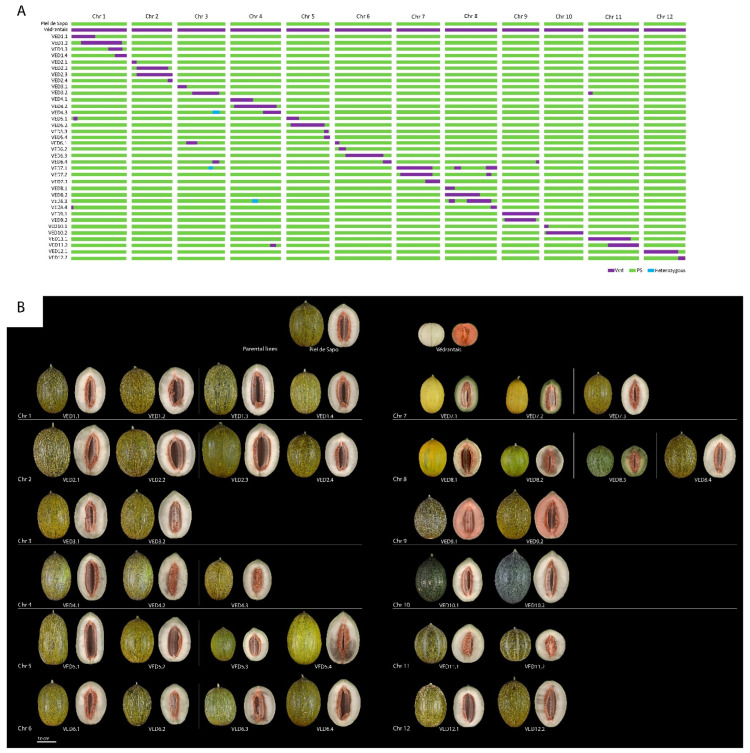
(**A**) Graphical representation of the genotypes of the introgression lines and the two parental lines. (**B**) External and internal appearance of each of the introgression lines and the parental lines.

**Figure 4 plants-11-03120-f004:**
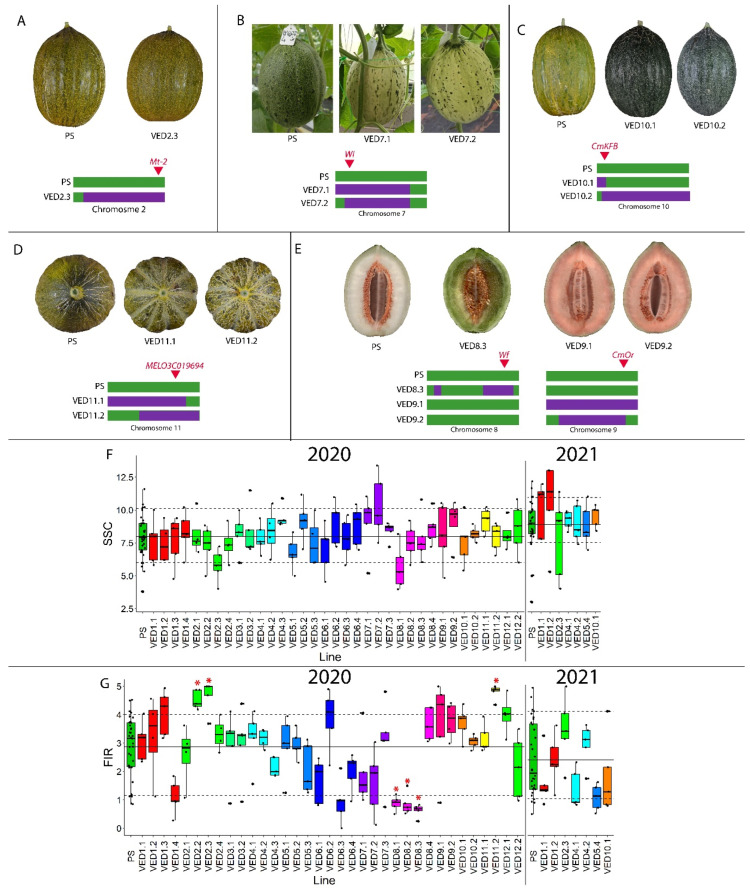
Fruit-quality-related phenotypes of the population in 2020 and 2021. (**A**) Presence of spots in the rind (mottled rind), with a representation of chromosome 2 and the reported causal gene. (**B**) Immature rind color, with a representation of chromosome 7 and the reported causal gene. (**C**) Yellowing of the rind, with a representation of chromosome 10 and the reported causal gene. (**D**) Presence of sutures in the rind, with a representation of chromosome 11 and the reported causal gene. (**E**) Flesh color, with a representation of chromosomes 8 and 9 and the reported causal genes. (**F**) Soluble solid content (SSC) in °Brix. (**G**) Flesh firmness (FIR) in kg cm^−2^. (**F**,**G**) Replicates are represented with dark circles; red asterisk indicates *p*-value < 0.05; different colors indicate different chromosomes.

**Figure 5 plants-11-03120-f005:**
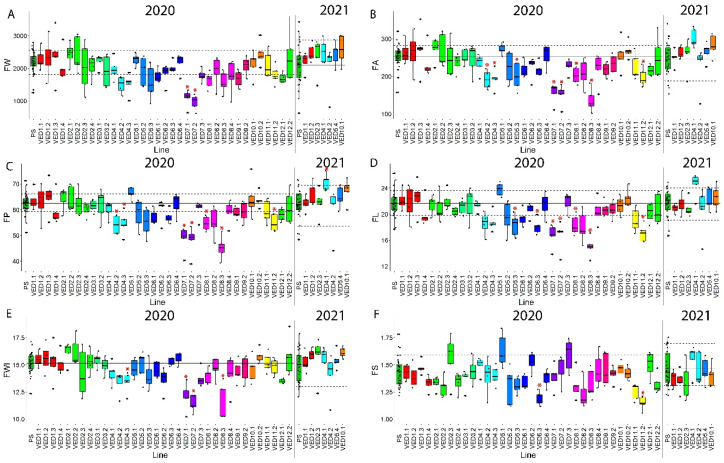
Fruit-morphology-related phenotypes of the population in both years 2020 and 2021. Replicates are represented with dark circles. Red asterisk indicates *p*-value < 0.05 and different colors indicate different chromosomes. (**A**) Fruit weight (FW) in grams. (**B**) Fruit area (FA) in squared centimeters (cm^2^). (**C**) Fruit perimeter (FP) in cm. (**D**) Fruit length (FL) in cm. (**E**) Fruit width (FWI) in cm. (**F**) Fruit shape (FS).

**Figure 6 plants-11-03120-f006:**
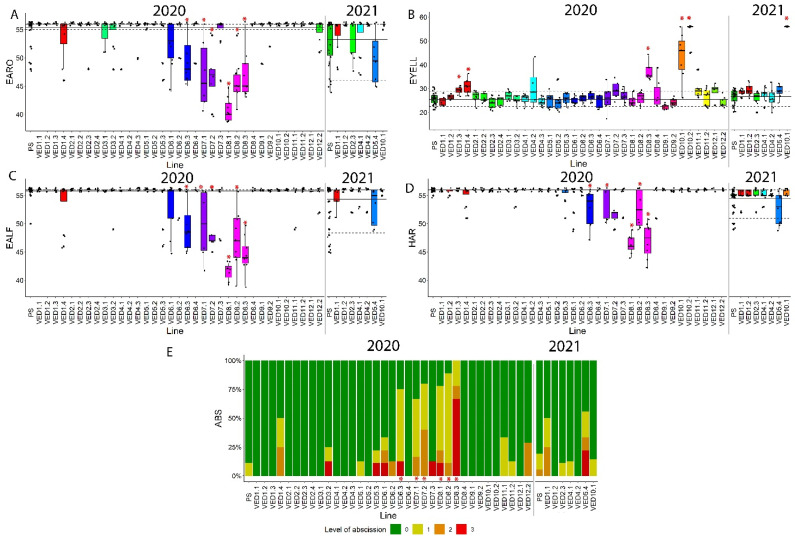
Ripening-related phenotypes of the population in 2020 and 2021. Replicates are represented with dark circles. Red asterisk indicates *p*-value < 0.05 and different colors indicate different chromosomes. (**A**) Earliness of aroma (EARO) in days after pollination (DAP). (**B**) Earliness of yellowing of the rind (EYELL) in DAP. (**C**) Abscission layer formation (EALF) in DAP. (**D**) Harvest date (HAR) in DAP. (**E**) Level of abscission (ABS) expressed as frequency (%).

**Table 1 plants-11-03120-t001:** Summary of the developed introgression lines population, with average introgression size per chromosome in physical and genetic distance.

Chr	Number of ILs	Introgression Size (bp)			Introgression Size * (cM)		
		Average	Min	Max	Average	Min	Max
1	4	15,012,079	7,083,403	23,611,517	49.5	37.7	58.3
2	4	12,741,987	2,293,836	23,903,255	57.1	14.7	100.6
3	2	11,857,082	6,062,824	17,651,340	53.2	36.8	69.7
4	3	17,188,567	11,842,362	24,802,312	69.6	65.5	75.1
5	4	9,353,072	2,799,579	22,919,604	45.5	24.9	63.3
6	4	9,574,343	2,430,563	27,849,153	38.2	21.9	66.7
7	3	19,189,571	8,232,872	25,979,214	81.8	67.5	100.8
8	4	12,919,496	1,895,546	25,002,865	47.8	12.9	92.1
9	2	23,045,254	20,847,233	25,243,276	85.6	61.7	109.5
10	2	15,007,760	5,108,239	24,907,280	70.3	54.0	86.6
11	2	25,057,113	20,321,128	29,793,099	89.6	54.0	99.4
12	2	13,781,830	4,579,847	22,983,813	51.9	47.1	56.8
Total	36	14,584,344	1,895,546	29,793,099	58.5	12.9	109.5

* Genetic distance based on the genetic map in Pereira et al. 2018 [23].

**Table 2 plants-11-03120-t002:** Summary of the QTLs mapped in the population for the quantitative traits.

Trait	QTL	IL	Year	Chr.	Genomic Interval (Mb)	IL Mean	PS Mean	% Difference ^a^	*p*-Value
FW	*FWQW7.1*	VED7.1	2020	7	2.62–20.73	1090.4	2175.6	−50.0	0.0131
		VED7.2	2020			963.6		−55.7	0.0148
FP	*FPQW4.1*	VED4.2	2020	4	22.48–27.25	54.17	62.48	−13.30	0.0183
		VED4.3	2020			54.58		−12.64	0.0259
	*FPQW8.1*	VED8.1	2020	8	0–6.89	54.65		−12.53	0.0053
	*FPQW7.1*	VED7.1	2020	7	2.62–20.73	48.66		−22.12	0.0004
		VED7.2	2020			48.08		−23.05	0.0002
	*FPQW8.2*	VED8.3	2020	8	31.12–34.62	45.2		−27.66	0.0002
	*FPQW11.1*	VED11.2	2020	11	29.79–34.46	54.89		−12.15	0.0071
	*FPQW4.2*	VED4.1	2021	4	0–14.92	71.52	62.2	14.98	0.0216
FA	*FAQW4.1*	VED4.2	2020	4	22.48–27.25	191.39	253.8	−24.59	0.0336
		VED4.3	2020			193.09		−23.92	0.0053
	*FAQW5.1*	VED5.3	2020		27.63–29.32	200.19		−21.12	0.0203
	*FAQW8.1*	VED8.1	2020	8	0–6.89	202.38		−20.26	0.0163
	*FAQW7.1*	VED7.1	2020	7	2.62–20.73	156.19		−38.46	0.0006
		VED7.2	2020			150.7		−40.62	0.0004
	*FAQW8.2*	VED8.3	2020	8	31.12–34.62	136.38		−46.26	0.0002
	*FAQW11.1*	VED11.2	2020	11	29.79–34.46	197.52		−22.17	0.0097
FL	*FLQW5.1*	VED5.3	2020	5	27.63–29.32	18.3	21.66	−15.51	0.0477
	*FLQW6.1*	VED6.3	2020	6	7.47–35.32	18.1		−16.44	0.0429
	*FLQW8.1*	VED8.1	2020	8	0–6.89	18.33		−15.37	0.0461
	*FLQW7.1*	VED7.1	2020	7	2.62–20.73	16.71		−22.85	0.0187
		VED7.2	2020			16.76		−22.62	0.0167
	*FLQW8.2*	VED8.3	2020	8	31.12–34.62	15.02		−30.66	0.0131
	*FLQW11.1*	VED11.2	2020	11	29.79–34.46	17.04		−21.33	0.0171
FWI	*FWIQW4.1*	VED4.3	2020	4	30.00–34.31	13.42	15.16	−11.48	0.0473
	*FWIQW7.1*	VED7.1	2020	7	2.62–20.73	12.07		−20.38	0.0171
		VED7.2	2020			11.41		−24.73	0.0130
	*FWIQW8.1*	VED8.3	2020	8	31.12–34.62	11.45		−24.47	0.0179
FS	*FSQW6.1*	VED6.3	2020	6	7.47–35.32	1.19	1.43	−16.78	0.0011
	*FSQW11.1*	VED11.2	2020	11	29.79–34.46	1.15		−19.58	<0.0001
FIR	*FIRQW2.1*	VED2.2	2020	2	3.16–24.77	4.52	2.86	58.04	0.0298
		VED2.3	2020			4.59		60.49	0.0475
	*FIRQW8.1*	VED8.1	2020	8	0–6.89	0.88		−69.23	0.0459
	*FIRQW8.2*	VED8.2	2020	8	6.89–14.98	0.85		−70.03	0.0334
	*FIRQW8.3*	VED8.3	2020	8	14.96–34.62	0.62		−78.32	0.0130
	*FIRQW11.1*	VED11.2	2020	11	29.79–34.46	4.8		67.83	0.0171
EARO	*EAROQW6.1*	VED6.3	2020	6	7.47–35.32	49.5	55.36	−10.59	0.0078
	*EAROQW8.1*	VED8.1	2020	8	0–6.89	40.89		−26.14	<0.0001
	*EAROQW8.2*	VED8.2	2020	8	6.89–14.98	46.1		−16.73	<0.0001
	*EAROQW8.3*	VED7.1	2020	8	31.12–34.62	47.17		−14.79	0.0063
		VED7.2	2020			46.8		−15.46	0.0007
		VED8.3	2020			46.9		−15.28	0.0001
EYELL	*EYELLQW1.1*	VED1.3	2020	1	29.95–34.47	29	26	0.12	0.0181
		VED1.4	2020			31		0.19	0.0076
	*EYELLQW8.1*	VED8.3	2020	8	25.00–29.53	37		0.42	0.0004
	*EYELLQW10.1*	VED10.1	2020	10	1.76–5.11	43		0.65	0.0083
		VED10.2	2020			54		1.08	0.0011
	*EYELLQW10.1*	VED10.1	2021	10	1.76–5.11	56	27	1.07	0.0002
EALF	*EALFQW6.1*	VED6.3	2020	6	7.47–35.32	49.38	55.81	−11.52	<0.0001
	*EALFQW8.1*	VED8.1	2020	8	2.63–6.89	41.43		−25.76	<0.0001
		VED8.3	2020			44.56		−20.16	<0.0001
	*EALFQW8.2*	VED8.2	2020	8	6.89–14.98	47.33		−15.19	<0.0001
	*EALFQW8.3*	VED7.1	2020	8	31.12–34.62	49.83		−10.71	0.0016
		VED7.2	2020			48.6		−12.92	0.0002
		VED8.3	2020			44.56		−20.16	<0.0001
ABS	*ABSQW6.1*	VED6.3	2020	6	7.47–35.32	1	0.11	0.89	0.0031
	*ABSQW8.1*	VED8.1	2020	8	2.63–6.89	1.11		1.00	0.0008
		VED8.3	2020			2.44		2.33	<0.0001
	*ABSQW8.2*	VED8.2	2020	8	6.89–14.98	1		0.89	<0.0001
	*ABSQW8.3*	VED7.1	2020	8	31.12–34.62	0.83		0.72	0.0369
		VED7.2	2020			1.2		1.09	0.0049
		VED8.3	2020			2.44		2.33	<0.0001
HAR	*HARQW6.1*	VED6.3	2020	6	7.47–35.32	52.75	55.97	−5.75	<0.0001
	*HARQW8.1*	VED8.1	2020	8	2.63–6.89	46.43		−17.05	<0.0001
		VED8.3	2020			47		−16.02	<0.0001
	*HARQW8.2*	VED8.2	2020	8	6.89–14.98	52.63		−5.97	0.0002
	*HARQW8.3*	VED7.2	2020	8	31.12–34.62	52		−7.09	<0.0001
		VED8.3	2020			47		−16.02	<0.0001

^a^ The background color gradient corresponds to the percentage of reduction (red) or increase (blue) in the IL phenotype when compared to PS.

**Table 3 plants-11-03120-t003:** Different phenotypes analyzed in this study, divided into three groups.

Group	Phenotype	Units *	Symbol
Fruit quality traits	Mottled rind		MOT
	Yellowing of the rind		YELL
	Presence of sutures		SUT
	Flesh color		FC
	External color of immature fruit		ECOL
	Soluble solid content	° Brix	SSC
	Firmness	kg cm^−2^	FIR
Fruit morphology traits	Fruit area	cm^2^	FA
	Fruit perimeter	cm	FP
	Fruit length	cm	FL
	Fruit width	cm	FWI
	Fruit shape		FS
	Fruit weight	g	FWI
Fruit ripening traits	Presence of aroma	DAP	EARO
	Earliness of yellowing of the rind	DAP	EYELL
	Abscission layer formation	DAP	EALF
	Level of abscission	**	ABS
	Harvest date	DAP	HAR

* DAP = days after pollination. ** Semi-quantitative trait, from 0 to 3.

## Data Availability

All data generated or analyzed during this study are included in this published article and its Appendix A.

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
