# Peer review of "Fruit Morphology and Ripening-Related QTLs in a Newly Developed Introgression Line Collection of the Elite Varieties ‘Védrantais’ and ‘Piel de Sapo’"

_plants, 2022, doi:10.3390/plants11223120_

Round 1

Reviewer 1 Report

The authors presented interesting results regarding the fruit morphology and ripening related QTLs detected by introgression lines of melon. The results are clear and the writings are readable. I cannot see other problems.

Reviewer 2 Report

Fruit Morphology and Ripening-Related QTLs in a Newly Developed Introgression Line Collection of the Elite Varieties ‘Védrantais’ and ‘Piel de Sapo’

In the manuscript "Fruit Morphology and Ripening-Related QTLs in a Newly Developed Introgression Line Collection of the Elite Varieties ‘Védrantais’ and ‘Piel de Sapo’", the authors developed an Introgression Line Collection and detected QTLs for important traits such as fruit ripening.

Introduction section

I suggest to the authors to remove the sentence “PS is a large, oval, white fleshed melon with non-climacteric ripening behavior, while Ved is a small, round, aromatic, orange-fleshed melon with typical climacteric behavior”, from the introduction section and include this sentence in the Materials and methods section.

Material and Methods section

The authors did not include in the material and methods section how the QTL mapping had been calculated. Which test have the authors applied for considering a QTL significant? Please include this information in the appropriated section.

Results section

ü  The authors indicate in the manuscript which parent revealed a higher fruit, amount of SSC and etc. I suggest that the authors include in the text the average value between parenthesis.

ü  The authors mentioned in the manuscript: “This high dependence on the environment can make it difficult to map QTLs related with this trait”. What are the authors' suggestions to overcome this problem?

ü  Please replace “When we looked at fruit shape” to “Concerning the fruit shape”

ü  Please replace “The biggest differences” to “Significant differences”

ü  The authors have written in the manuscript: “While Ved ripened at around 35 days after pollination in both years, PS ripened around 20 days later, generally without producing aroma or abscission layer (Figure 1G, H, I) (Table S1)”. Are these results in agreement with previous studies?

ü  The authors included a table (table 1) comparing introgression size reported in this study with the previously reported by Pereira et al. 2018. However, the authors did not discuss the results in the text of the manuscript.

ü  I suggest that the authors include the unit in which each trait was measured in the supplementary tables and graphs.

ü  The authors mentioned “During the phenotyping, several fruit quality traits were analyzed”. I suggest that the authors write in the text the traits analyzed.

ü  The authors mentioned “the population was genotyped using the complete set of SNPs with some modifications of SNPs that did not work properly”. I suggest that the authors describe the modifications.

ü  The authors highlighted in the text: “A consensus QTL in this region has been previously reported affecting fruit width in a RIL population [47]”. Is this QTL reported in the same region? Are the traits related to fruit size correlated with fruit weight?

ü  The authors mentioned a QTL cluster detected in 2020. However, this QTL was not detected in 2021. What are the possible reasons?    

In my view, the results section, especially the discussion could be further improved. The authors can improve the structure of the text, emphasizing the importance of the traits analyzed, which are the QTLs reported for the first time, how the results reported in this work impact the future studies in the melon breeding programs, the next steps and the strategies for overcoming the problems such as the traits that are extremely affected by the environment and unstable QTLs observed in this study. In addition, the conclusion should be rewritten and improved.

Reviewer 3 Report

Understanding the genetic basis of diverse fruit morphology and ripening characteristics in melons is of great importance. In this study, the authors constructed an IL population through more than three rounds of backcrossing with ‘Piel de Sapo’ (PS) as the recurrent parent and ‘Védrantais’ (Ved) as the donor parent. These two elite cultivars were chosen due to their significant differences in fruit morphology and ripening characteristics.

I have several concerns about this research. First, the statistical power of the loci. As shown in figure 4A, there is only one plant carrying the Ved genome. As they indicated in the method, “Wilcoxon signed-rank test was used to compare each IL with PS”,  which is not a statistic for this locus. I prefer to see some statistics that describe the % of variance explained as a whole instead of a detailed phenotypic reading for each individual like in Figure4. And also, the author should include the comparison in 2021 no matter if the result is significant or not.

When it is possible, I would highly suggest the authors try to increase the marker density for this population. A panel with 96 SNPs is fairly low.

Minor issues:

In Figure 2, I would suggest the author plot the average of the recurrent parent.

In Figure 4, I suggest highlighting the two groups in which you made the comparison.

Round 2

Reviewer 3 Report

The authors have addressed all of my concerns. I have no further questions.